# Evaluation of CpG-ODN-Adjuvanted *Toxoplasma gondii* Virus-Like Particle Vaccine upon One, Two, and Three Immunizations

**DOI:** 10.3390/pharmaceutics12100989

**Published:** 2020-10-19

**Authors:** Hae-Ji Kang, Ki-Back Chu, Min-Ju Kim, Hyunwoo Park, Hui Jin, Su-Hwa Lee, Eun-Kyung Moon, Fu-Shi Quan

**Affiliations:** 1Department of Biomedical Science, Graduate School, Kyung Hee University, Seoul 02447, Korea; heajik0514@khu.ac.kr (H.-J.K.); Kbchu@khu.ac.kr (K.-B.C.); mj16441@khu.ac.kr (M.-J.K.); 2Health Park Co., Ltd., Seoul 06627, Korea; hwpark75@gmail.com (H.P.); kimhwi83@gmail.com (H.J.); 3Department of Medical Zoology, Kyung Hee University School of Medicine, Seoul 02447, Korea; dltnghk228@khu.ac.kr (S.-H.L.); ekmoon@khu.ac.kr (E.-K.M.); 4Department of Medical Research Center for Bioreaction to Reactive Oxygen Species and Biomedical Science Institute, School of Medicine, Graduate School, Kyung Hee University, Seoul 02447, Korea

**Keywords:** *Toxoplasma gondii*, virus-like particle, vaccine, CpG-ODN

## Abstract

Successful vaccines against specific pathogens often require multiple immunizations and adjuvant usage. Yet, assessing the protective efficacy of different immunization regimens with adjuvanted *Toxoplasma gondii* vaccines remains elusive. In this study, we investigated the vaccine efficacy induced by CpG-ODN-adjuvanted *T. gondii* virus-like particles (VLPs) after challenge infection with *T. gondii* (ME49) in mice (BALB/c) upon one, two, and three immunizations. Immunization with adjuvanted *T. gondii* VLPs induced higher levels of *T. gondii*-specific IgG and/or IgA antibody responses, germinal center (GC) B cells, total B cells, and CD4^+^ and CD8^+^ T cells compared with unadjuvanted VLPs. Increasing the number of immunizations was strongly correlated with enhanced protective immunity against *T. gondii* in mice, with the highest protection being demonstrated in mice thrice-immunized with either adjuvanted *T. gondii* VLPs or VLPs alone. Notably, lesser bodyweight reductions and cerebral cyst counts were observed in mice receiving multiple immunizations with the adjuvanted VLPs, thereby confirming the effectiveness of adjuvanted boost immunizations. These results demonstrated that multiple immunizations with *T. gondii* VLPs is an effective approach, and the CpG-ODN can be developed as an effective adjuvant for *T. gondii* VLP vaccines.

## 1. Introduction

*Toxoplasma gondii* is an obligate intracellular parasite widely distributed across the globe and infects a vast array of mammals including humans [1,2,3]. *T. gondii* infection in pregnant women and acquired immune deficiency syndrome (AIDS) patients can have severe consequences such as spontaneous abortion and encephalitis [4,5,6]. To date, clinical *T. gondii* vaccines remain commercially unavailable. While therapeutic interventions are possible, their applications are hindered by toxicity and other side effects [7,8]. To address these limitations, multitudes of *T. gondii* vaccine studies are currently being conducted using DNA, protein subunit, inactivated, and attenuated vaccines to develop an effective toxoplasmosis vaccine [9].

Conflicting protective efficacy results have been reported through numerous vaccine studies. All of the mice immunized with the DNA vaccine encoding the *T. gondii* surface antigen 1 (SAG1) survived upon *T. gondii* (ME49) challenge infection [10], whereas none of the mice immunized with the DNA vaccine expressing *T. gondii* superoxide dismutase (SOD) survived [11]. Survival discrepancies were also observed from mice immunized with various subunit vaccines before challenge infection with *T. gondii* ME49 [12,13,14]. In contrast to the DNA or recombinant subunit vaccines, immunizing the mice with attenuated *T. gondii* ensured that all of the immunized mice survived following challenge infection with a lethal dose of ME49 [15,16]. Though the protective efficacies of the live attenuated vaccines appear promising, the safety aspects of these vaccines are of concern since attenuated *T. gondii* can revert to the highly pathogenic wild type [14]. As a safer alternative, we generated several virus-like particle (VLP) *T. gondii* vaccines conferring 100% protection against a lethal dose of *T. gondii* ME49 strain in mice [17,18,19,20]. Although all of the immunized mice survived in our previous studies, incomplete removal of cerebral cysts and bodyweight loss upon challenge infection from these mice indicated that further improvements to the VLP vaccines are needed to minimize disease manifestation.

Synthetic oligodeoxynucleotides containing unmethylated CpG motifs (CpG-ODNs) have been shown to act as immunologic adjuvants in mice, which enhances humoral and cellular responses induced by co-administered vaccines [21,22]. C-Class CpG-ODNs induce strong interferon-alpha (IFN-α) production from the plasmacytoid dendritic cell (pDC) as well as B-cell stimulation [22]. To this extent, combining the highly immunogenic *T. gondii* VLP vaccines with CpG-ODN adjuvants could confer enhanced protection with close to no symptoms. Multiple immunizations are of utmost importance for adequate adaptive immunity induction. An assessment of the protective efficacy of different immunization regimens with adjuvanted *T. gondii* VLPs vaccines is urgently needed.

In the current study, mice were intranasally immunized with the CpG-ODN-adjuvanted VLPs once, twice, or thrice and the resulting immune responses were assessed. We found that the highest protection was found from mice thrice-immunized with adjuvanted *T. gondii* VLPs. Our findings highlight the importance of this multi-immunization approach and adjuvant CpG usage in eliciting potent antibody responses and protection.

## 2. Materials and Methods

### 2.1. Mice and Parasite

Female, 6–8-week-old, BALB/c mice were purchased from NARA Biotech (Seoul, Korea) and maintained in the animal facility at Kyung Hee University. All animal experiments were performed following the institutional animal care and use institutional animal care and use committee (IACUC) guidelines (permit number: KHUASP (SE)-18-050). *T. gondii* ME49 and RH strains were maintained and used for experimental infections as previously described [23,24].

### 2.2. VLP Vaccine and Reagents

TG146 VLP vaccine expressing *T. gondii* IMC, ROP18, and MIC8 was produced in insect cells as described previously [18,20]. The multi-antigenic TG146 VLPs were aliquoted and stored at −80 °C until use. Lyophilized CpG-ODN (ODN 2395) was purchased from InvivoGen (San Diego, CA, USA) and reconstituted using ultra-pure water following the manufacturer’s protocol. CpG-ODN adjuvants were aliquoted and stored at −20 °C until use.

### 2.3. Immunization and T. gondii ME49 Infection

To determine the efficacy of the CpG-ODN adjuvant against *T. gondii* ME49, BALB/c mice (*n* = 6 per group) were intranasally immunized with *T. gondii* VLPs (100 μg) alone or *T. gondii* VLPs + CpG ODN (100 μg + 5 μg). After prime immunization, mice were immunized with the same doses of either *T. gondii* VLPs alone or *T. gondii* VLPs + CpG ODN at 4-week intervals for second and third immunizations. Immune sera were collected at 1 and 4 weeks after each immunization. At 4 weeks after the final immunization, mice immunized once, twice, or thrice were orally infected with a lethal dose of *T. gondii* ME49 (One + Cha, Two + Cha, Three + Cha). After challenge infection, mice were monitored for 35 days to record body weight changes and survival rates. To compare the protective efficacy and immunological responses upon *T. gondii* ME49 infection, mice were euthanized 35 days post-infection (dpi) and mucosal antibodies, spleen, mesenteric lymph nodes (MLNs), and brain tissues were collected.

### 2.4. Sample Preparation

To determine mucosal immune responses, mouse mucosal samples (feces, intestine, urine, and vaginal samples) and brains of the immunized mice were collected at day 35 after *T. gondii* ME49 infection. Ten pieces of feces were collected from each mouse and 100 μL of phosphate-buffered saline (PBS) was added per 0.1 g of feces. For intestinal sample preparation, a 5 cm long duodenum tissue starting from the pyloric sphincter was acquired from each mouse. Duodenum was longitudinally incised and resuspended in 800 μL PBS. Urine samples from mice undergoing euthanasia were carefully collected for use. Vaginal secretions were collected by washing the vagina twice with 200 μL of PBS using a pipette. All mucosal samples were incubated for 1 h at 37 °C, then centrifuged for 10 min at 5000 rpm. Following the centrifugation process, supernatants were collected for the antibody assay. Brain tissues were homogenized using a syringe in 400 μL of PBS and centrifuged at 8000 rpm. Supernatants were used to measure cerebral antibody and cytokine responses. Remaining brain tissues were used to quantify brain cysts and their sizes as previously described [25,26].

### 2.5. ELISA for Antibody Responses and Cytokine Levels

*T. gondii*-specific antibody and cytokine levels were determined by ELISA as reported previously [18,26]. Diluted immune sera (1:100), urine (1:3), vaginal samples (1:3), fecal supernatants (1:3), intestine samples (1:100), and brain supernatants (undiluted) were used as primary antibodies. Briefly, 96-well plates were coated with *T. gondii* RH (2 μg/mL) and blocked prior to incubation with the primary antibodies listed above. Anti-mouse IgG and IgA secondary antibodies conjugated with horseradish peroxidase (HRP) were used to detect the antigen-specific responses. The production of pro-inflammatory cytokines interferon-gamma (IFN-γ) and interleukin-6 (IL-6) were measured from the brain supernatants using the cytokine ELISA kits (BD Biosciences, San Jose, CA, USA).

### 2.6. Flow Cytometry

Single cell suspensions of splenocytes and MLN cells acquired from mice were used for fluorescence-activated cell sorting (FACS) analysis as previously described [26]. Cells were unstimulated or stimulated with *T. gondii* RH antigens (1 μg/mL) for 2 h, treated with CD16/32 antibody for blocking Fc receptors, and subsequently incubated with fluorophore-labeled antibodies specific for anti-mouse CD3 (PE-Cy7), CD4 (FITC), CD8 (PE), GL7 (PE), B220 (FITC), IgD (PE), and CD19 (PE-Cy7). All antibodies were purchased from Invitrogen (Carlsbad, CA, USA) and BD Biosciences (San Diego, CA, USA). Cell acquisition and data analysis were performed using BD Accuri C6.

### 2.7. Statistical Analysis

GraphPad Prism version 5 software was used for statistical analysis. Data sets were presented as mean ± SD. One-way ANOVA with Tukey’s post hoc test or two-way ANOVA with Bonferroni’s post hoc test was used to test for statistical significance between each group. *p* values of less than 0.05 were considered statistically significant (* *p* < 0.05, ** *p* < 0.01, *** *p* < 0.001).

## 3. Results

### 3.1. T. gondii-Specific IgG and IgA Antibody Responses in Sera

To determine whether the CpG-ODN adjuvant can enhance the efficacy of the *T. gondii* VLP vaccine after multiple immunizations, mice were immunized with either *T. gondii* VLPs alone or CpG-ODN-adjuvanted *T. gondii* VLPs for one, two, or three times. As expected, CpG-ODN-adjuvanted *T. gondii* VLP vaccination induced higher levels of *T. gondii*-specific IgG (Figure 1A) and IgA (Figure 1B) antibodies compared with unadjuvanted *T. gondii* VLPs upon one, two, and three immunizations. A positive correlation between antibody inductions and the number of immunizations was observed as anticipated. The highest IgG and IgA responses were observed from the sera of mice immunized thrice, while the lowest antibody inductions were found in mice immunized once. Notably, regardless of the vaccination schedule, the presence of CpG-ODN adjuvants effectively bolstered the serum antibody responses.

### 3.2. T. gondii—Specific IgG and IgA Antibody Responses in Mucosal Samples

Mucosal antibody responses from naive, naive-challenged (Naïve + Cha), and immunized mice were determined using feces (Figure 2A,B), intestines (Figure 2C,D), urine (Figure 2E,F), and vaginal secretions (Figure 2G,H). Immunization with the adjuvanted *T. gondii* VLP vaccine elicited higher levels of *T. gondii*-specific IgG and IgA antibody responses compared with unadjuvanted *T. gondii* VLPs, indicating adjuvant effects in inducing IgG and IgA antibody responses in all mucosal samples. The highest IgG and IgA antibody responses were elicited upon three immunizations in intestine samples after challenge infection with *T. gondii*.

### 3.3. Immunization with T. gondii VLPs Alone or Adjuvanted VLPs Induces CD4^+^ and CD8^+^ T-Cell Expansion

At 35 dpi, spleens and MLNs were collected from mice immunized once, twice, or thrice with either unadjuvanted *T. gondii* VLPs or CpG-ODN-adjuvanted *T. gondii* VLPs to determine the effect of adjuvants on CD4^+^ and CD8^+^ T-cell frequency. To identify the cells and quantify the percentage, CD4^+^ T cells, CD8^+^ T cells, and germinal center B cells in spleen and MLNs were gated as illustrated below (Figure 3). CpG-ODN-adjuvanted *T. gondii* VLP vaccination showed high levels of CD4^+^ (Figure 4A,C) and CD8^+^ (Figure 4B,D) T-cell frequencies in spleen and MLNs compared with *T. gondii* VLPs alone, indicating that the adjuvant CpG-ODN played a crucial role in inducing CD4^+^ and CD8^+^ T-cell responses. Enhancing T-cell expansion in both spleen and MLNs was dependent on adjuvant usage and the number of immunizations.

### 3.4. CpG-ODN-Adjuvanted VLP Vaccination Induced Germinal Center (GC) B-Cell and B-Cell Responses

CpG-ODN adjuvants are known to be potent immune stimulators that activate GC B cells and B cells. To evaluate the CpG-ODN adjuvant effects on GC B-cell and B-cell response inductions, murine splenocytes and MLN cells were harvested at 35 dpi with *T. gondii* ME49. As seen in Figure 5, adjuvanted VLP vaccination induced higher levels of GC B cells (Figure 5A,C) and B cells (Figure 5B,D) in spleen and MLNs. Induced GC B-cell and B-cell responses were proportional to the number of immunizations, with mice receiving three immunizations showing better responses than mice immunized once or twice. GC B-cell induction was the highest in the spleen (12%) and B cells were induced to the highest extent in MLNs (33.8%), both of which were observed in mice immunized thrice with the adjuvanted VLPs.

### 3.5. Antibody and Pro-Inflammatory Cytokine Responses in the Brain

Following *T. gondii* VLP immunization with or without adjuvants, brains of mice were collected at 35 dpi to evaluate antibody responses and pro-inflammatory cytokine production. Adjuvanted *T. gondii* VLPs elicited greater *T. gondii*-specific IgG (Figure 6A) and IgA (Figure 6B) responses in the brain in comparison with unadjuvanted *T. gondii* VLPs, with the highest antibody inductions occurring in mice immunized thrice. Immunization with the adjuvanted *T. gondii* VLPs lessened the production of pro-inflammatory cytokines IFN-γ (Figure 6C) and IL-6 (Figure 6D) compared with unadjuvanted *T. gondii* VLPs alone. *T. gondii* infection in unimmunized mice resulted in the highest production of these pro-inflammatory cytokines, while a marked decline in their levels became evident with the increasing number of immunizations.

### 3.6. Adjuvanted T. gondii VLP Vaccination Significantly Reduced Cyst Counts upon Challenge Infection with T. gondii ME49

To further assess the effects of *T. gondii* VLPs formulated with CpG-ODN adjuvants, key parameters indicating protection such as cyst size and their counts in the brain were measured. Different groups of mice received a different number of immunizations with or without CpG-ODN adjuvants. Upon challenge infection, mice were sacrificed at 35 dpi to evaluate reductions in brain cyst size and their counts under the microscope. Cyst sizes observed from the brains of mice immunized once, twice, or thrice were 35.1 μm, 35.6 μm, and 30.3 μm, respectively (Figure 7A). Compared with the cysts observed from unimmunized control (44.8 μm), significant reductions in cyst sizes were detected from immunized mice which were unaffected by adjuvants. Cyst formation was suppressed to a greater extent in mice immunized with *T. gondii* VLPs formulated with CpG-ODN adjuvants than those receiving *T. gondii* VLPs alone (Figure 7B). Cyst counts from immunization with *T. gondii* VLPs alone once, twice, or thrice were 17,100, 5870, and 2010, respectively. Immunization with adjuvant-formulated VLPs once, twice, or thrice further lessened these counts to 13,400, 3750, and 807, respectively. These results indicated that adjuvant usage and multiple immunizations have a profound effect on reducing total cyst counts.

### 3.7. Protective Efficacy Induced by T. gondii VLPs with or without Adjuvant CpG-ODN

To determine the effects of adjuvants and multiple immunizations on improving protective efficacy, immunized mice were challenged with a lethal dose of *T. gondii* ME49, 4 weeks after the last immunization. Bodyweight changes (Figure 8A,C,E) and survival rates (Figure 8B,D,F) were monitored for 35 days after challenge infection. Naive mice underwent gradual bodyweight loss, eventually reaching the humane intervention point by 35 dpi. On the contrary, all of the immunized mice survived and were subjected to lesser weight loss than the unimmunized control. Notably, increasing the number of immunizations lessened the bodyweight loss and this was further reduced through adjuvant incorporation. Mice immunized thrice with VLPs alone or adjuvanted VLPs experienced 7.8% and 3.6% bodyweight loss, respectively. Nearly two-fold differences in the percentage of bodyweight loss were observed between mice immunized twice and thrice, indicating the importance of adjuvant and multiple immunizations in inducing protection.

## 4. Discussion and Conclusions

Successful vaccine development requires adjuvants that can induce enhanced humoral and cellular immunity [27,28]. In our previous study, immunization with the *T. gondii* VLPs alone following the prime–boost regimen successfully induced protection by eliciting both humoral and cellular immunity [17,18,26,29]. We hypothesized that introducing the CpG-ODN adjuvant into *T. gondii* VLP vaccine formulation could induce enhanced protective immune responses, thereby improving the overall vaccine efficacy. Thus, in this study, the effect of the CpG-ODN adjuvant was evaluated using three different immunization regimens in mice, and the resulting protection against *T. gondii* infection was compared.

The toll-like receptor 9 (TLR9) agonist C-class CpG-ODN embodies the features of both classes A and B, enabling potent antibody induction and cytokine secretion by activating dendritic cells (DCs) and B cells [30,31]. The absence of adverse side effects even in higher eukaryotic organisms such as primates further validates its use as a vaccine adjuvant. Evidently, CpG-ODN-adjuvanted recombinant hepatitis B virus vaccine has recently been approved for clinical use [32]. Thus far, several studies have reported the protective efficacies of protein and DNA vaccines formulated with CpG-ODN adjuvants [33,34,35,36]. In our current study, we found that the efficacy of the CpG-ODN-adjuvanted *T. gondii* VLP vaccine was significantly enhanced compared with unadjuvanted VLPs after one, two, and three immunizations.

Immunization regimen is another important aspect contributing to vaccine efficacy. The prime–boost strategy has been used in our previous studies where significantly higher levels of IgG and/or IgA antibody responses were elicited after boost compared to the prime, resulting in a significant reduction of cerebral cyst counts and body weight loss upon challenge infection with *T. gondii* ME49 [17,18,26,29]. Recently, a clinical study assessing the efficacy of a pertussis vaccine reported that enhanced vaccine effectiveness was observed from subjects receiving two vaccine boosters compared with those receiving a single booster vaccination [37]. In our current study, increased number of immunizations using CpG-ODN-adjuvanted *T. gondii* VLPs significantly increased the protective humoral and cellular immune responses.

Intranasal (IN) immunization is known to elicit mucosa immunity as well as systemic immunity [38]. Mucosal immunity induction is particularly important since *T. gondii* transmission mainly occurs through oral ingestion of the pathogen [39,40]. The CpG-ODN adjuvant has been proved to be a promising mucosal adjuvant [41,42] and induces better protective mucosal immunity after IN immunization [43,44]. In the current study, upon intranasal immunization, mucosal IgG and IgA antibodies were found to be detected in various mucosal sites. Among them, the highest parasite-specific IgG and IgA antibody responses were detected from intestinal samples of mice immunized three times with the adjuvanted VLPs. Secretions of these antibodies are crucial for protection, as they often serve as the first line of defense against pathogens [45,46]. In line with this notion, the least number of brain cysts were found in mice receiving three immunizations with the adjuvanted VLPs, which possessed the highest level of mucosal antibodies.

Spleen and MLNs are secondary lymphoid organs that activate lymphocytes and initiate an adaptive immune response [47]. The adaptive immunity induced by vaccination depends on the presence of T cells and B cells. In our current study, *T. gondii*-specific CD4^+^ T cells, CD8^+^ T cells, GC B cells, and B-cell responses in both spleen and MLNs were found to be higher in adjuvanted VLP vaccination compared with unadjuvanted VLPs, which were significantly enhanced after two or three immunizations.

Pro-inflammatory cytokines induced by *T. gondii* infection tend to exacerbate the disease itself or symptoms associated with the disease [25,48,49]. After infection with *T. gondii* ME49, pro-inflammatory cytokines appear most frequently in the brain. In our current study, adjuvanted VLP vaccination showed significant reductions of pro-inflammatory cytokines IFN-γ and IL-6 in the brain compared with unadjuvanted VLPs. The induction of inflammatory cytokines correlated with bodyweight reduction and cyst counts. The lowest cytokine levels were detected from mice immunized thrice with the adjuvanted VLPs, which correspondingly had the least bodyweight loss and cyst counts. Lesser number of immunizations resulted in greater pro-inflammatory cytokine production, cyst counts, and bodyweight loss as indicated by mice immunized once or twice.

In summary, our results demonstrated that the adjuvants and multiple immunizations are necessary for developing an effective *T. gondii* VLP vaccine. Multiple immunizations with the CpG-ODN-adjuvanted VLPs conferred protection against a lethal dose of *T. gondii* ME49 by eliciting strongly enhanced cellular and humoral immunity.

## Figures and Tables

**Figure 1 pharmaceutics-12-00989-f001:**
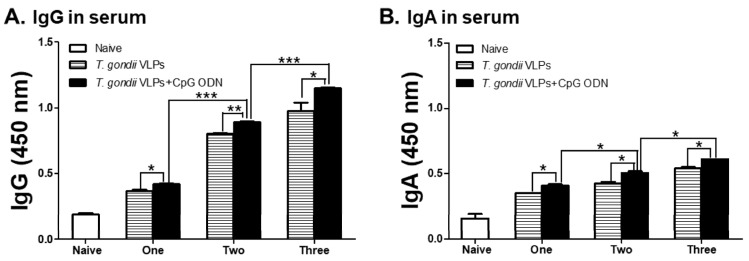
Immunization with CpG-ODN-adjuvanted *Toxoplasma gondii* virus-like particles (VLPs) enhances *T. gondii*-specific IgG (**A**) and IgA (**B**) antibody production in the sera upon one, two, and three immunizations. Groups of mice (*n* = 6 per group) were intranasally immunized one, two, or three times with *T. gondii* VLPs alone or CpG-ODN-adjuvanted *T. gondii* VLP vaccine. Booster vaccinations were provided at 4-week intervals. Mouse sera were collected at 4 weeks after each immunization to detect the levels of IgG (**A**) and IgA (**B**) antibodies by ELISA. Results are presented as mean ± SD and statistical significances were denoted using asterisks (* *p* < 0.05, ** *p* < 0.01, and *** *p* < 0.001).

**Figure 2 pharmaceutics-12-00989-f002:**
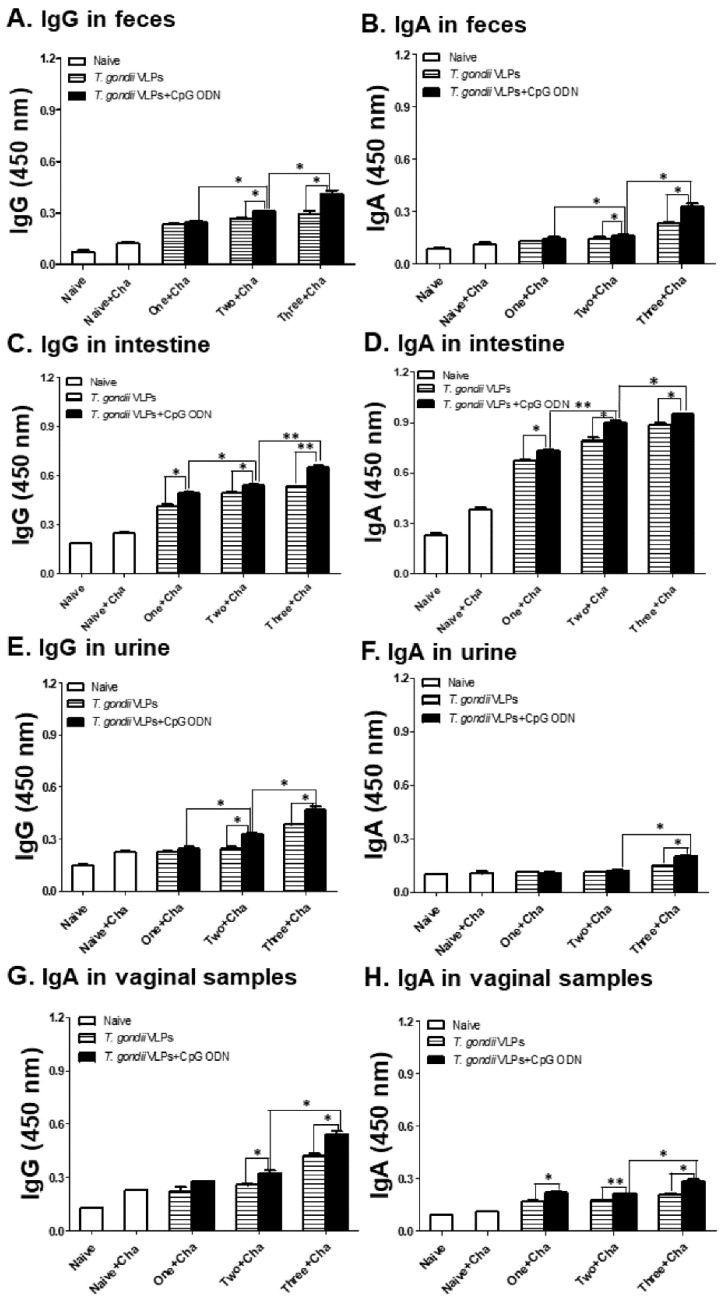
IgG and IgA antibody responses in mucosal samples after immunization with *T. gondii* VLPs alone or adjuvanted VLPs. Mice (*n* = 6 per group) were intranasally immunized with *T. gondii* VLPs alone or adjuvanted *T. gondii* VLP vaccine one, two, or three times. IgG and IgA mucosal antibody responses from feces (**A**,**B**), intestines (**C**,**D**), urine (**E**,**F**), and vaginal samples (**G**,**H**) were determined post-challenge with *T. gondii* ME49. Results are presented as mean ± SD and statistical significances were denoted using asterisks (* *p* < 0.05 and ** *p* < 0.01).

**Figure 3 pharmaceutics-12-00989-f003:**
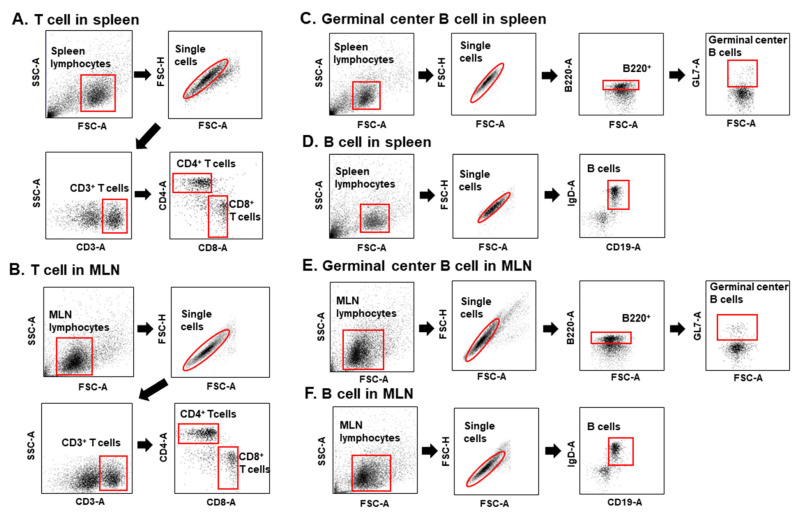
The gating strategy of CD4^+^ T cells, CD8^+^ T cells, and B cells in spleen and mesenteric lymph nodes (MLNs). Spleen and MLN cells from mice (*n* = 6) were collected at 35 dpi upon challenge infection with *T. gondii* ME49 and stained with phenotype-specific antibodies (CD3, CD4, CD8, B220, and GL7). The gating strategies depicted above were applied to identify CD4^+^ and CD8^+^ T cells (**A**,**B**), germinal center B cells, and B cells from both spleen and MLNs (**C**–**F**).

**Figure 4 pharmaceutics-12-00989-f004:**
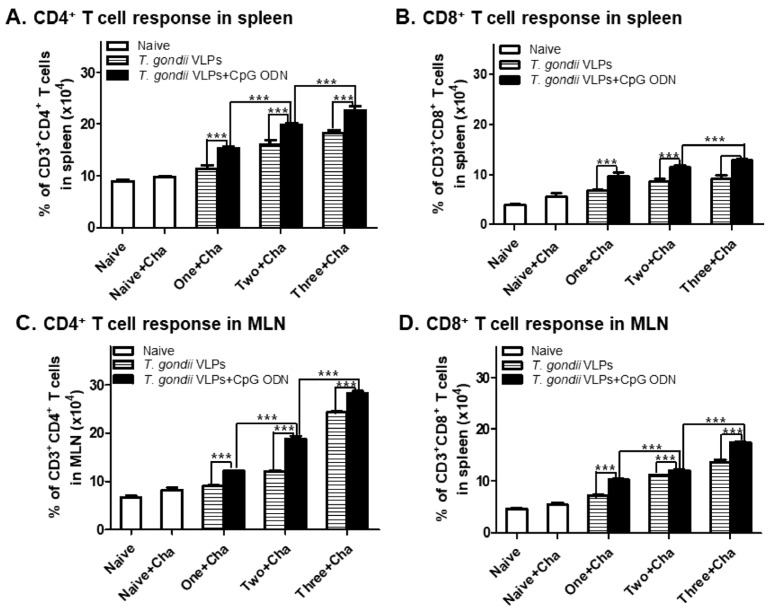
Immunization with CpG-ODN-adjuvanted *T. gondii* VLPs enhances CD4^+^ (**A**,**C**) and CD8^+^ (**B**,**D**) T-cell frequencies. BALB/c mice (*n* = 6 per group) immunized one, two, or three times were challenge-infected with *T. gondii* ME49, 4 weeks after the last immunization. Spleen and MLN tissues were harvested at 35 dpi and then cell phenotypes were determined by flow cytometry with or without *T. gondii* antigen stimulation. Results are presented as mean ± SD and statistical significances were denoted using asterisks (*** *p* < 0.001).

**Figure 5 pharmaceutics-12-00989-f005:**
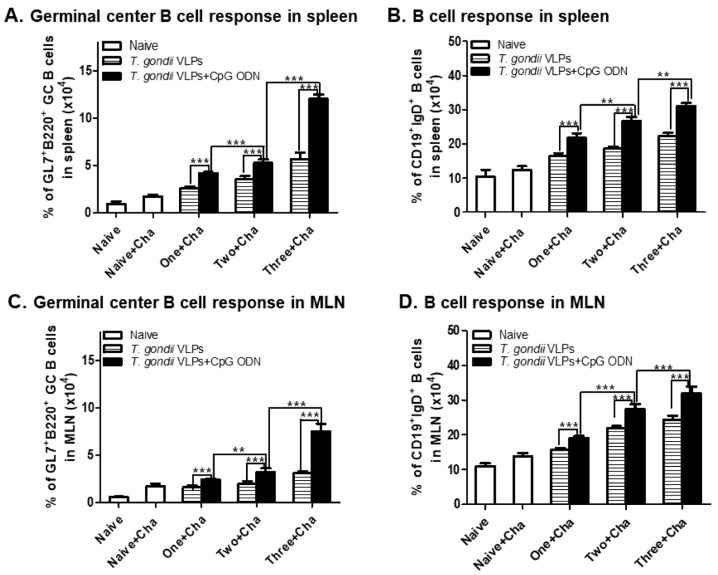
Immunization with the adjuvanted *T. gondii* VLPs enhances antigen-specific B-cell responses. Immunized mice (*n* = 6 per group) were infected with *T. gondii* ME49, 4 weeks after the last immunization. Spleen and MLN tissues were harvested at 35 dpi and then cell phenotypes were determined by flow cytometry. The percentage of GC B (**A**,**C**) cells or B cells (**B**,**D**) was quantified by subtracting unstimulated cells from stimulated cells. Results are presented as mean ± SD and statistical significances were denoted using asterisks (** *p* < 0.01 and *** *p* < 0.001).

**Figure 6 pharmaceutics-12-00989-f006:**
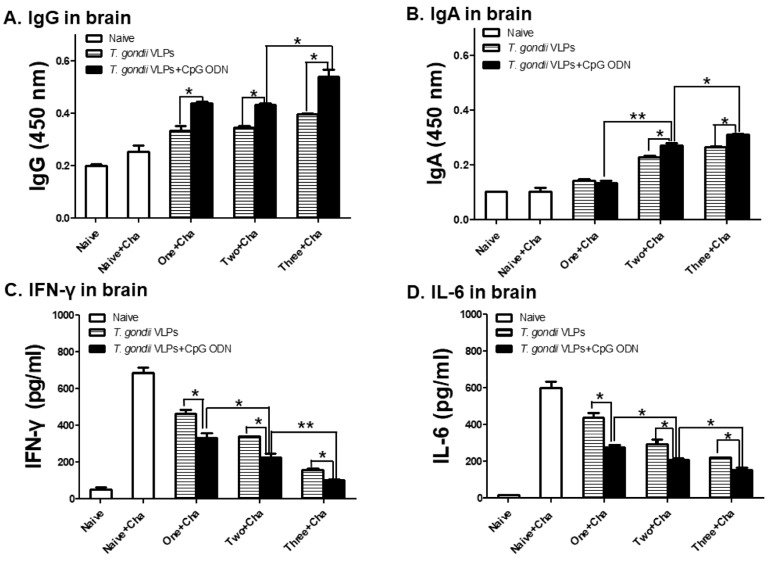
Antibody responses and pro-inflammatory cytokine responses in the brain upon one, two, and three immunizations. Mice (*n* = 6 per group) were immunized once, twice, or thrice with either adjuvanted or unadjuvanted *T. gondii* VLPs, and their brains were collected at 35 dpi with *T. gondii*. Brain supernatants were prepared and pro-inflammatory cytokines were measured by ELISA. Results (**A**–**D**) are presented as mean ± SD and statistical significances were denoted using asterisks (* *p* < 0.05 and ** *p* < 0.01).

**Figure 7 pharmaceutics-12-00989-f007:**
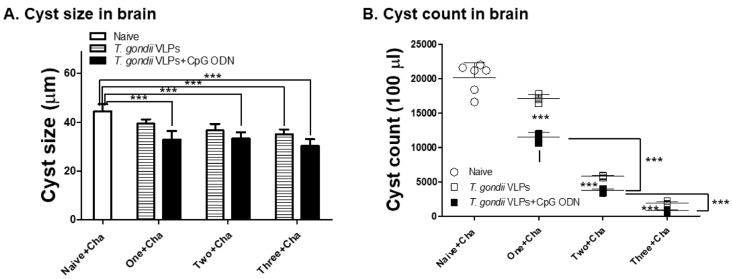
CpG-ODN-adjuvanted *T. gondii* VLP immunization reduced *T. gondii* cyst sizes and counts. Immunized mice (*n* = 6 per group) were challenge-infected 4 weeks after the last immunization and sacrificed 35 dpi. The cyst sizes (**A**) and counts (**B**) of *T. gondii* in the brains were measured. Results are presented as mean ± SD and statistical significances were denoted using asterisks (*** *p* < 0.001).

**Figure 8 pharmaceutics-12-00989-f008:**
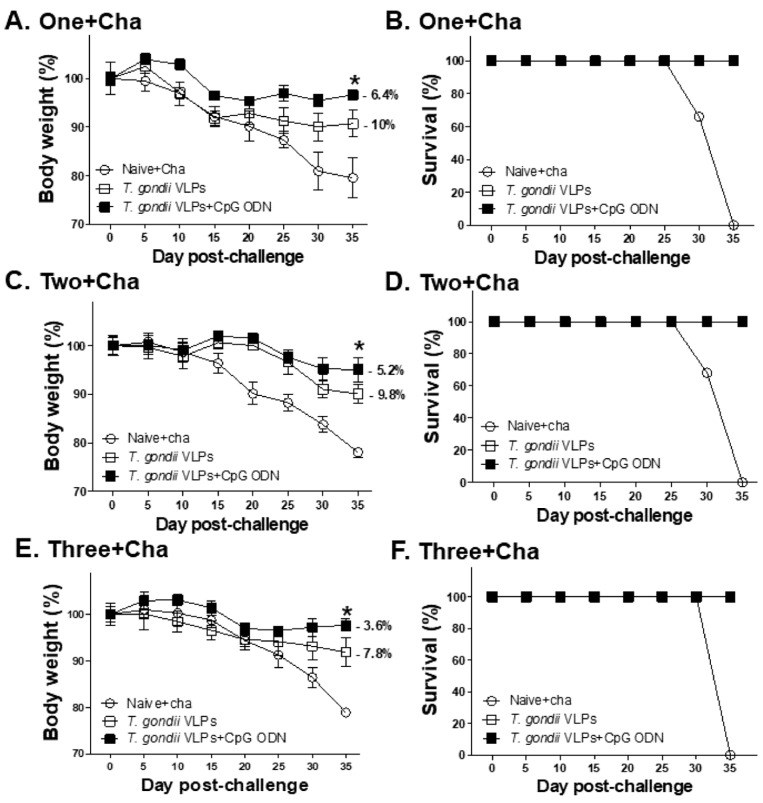
Protective efficacy induced by *T. gondii* VLPs alone and adjuvanted *T. gondii* VLPs following multiple immunizations. Immunized mice (*n* = 6 per group) were challenge-infected after the final immunization and monitored for 35 days. Bodyweight reductions (**A**,**C**,**E**) and survival rates (**B**,**D**,**F**) from mice immunized once, twice, or thrice were measured during this period. Results are presented as mean ± SD and statistical significances were denoted using asterisks (* *p* < 0.05).

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
