# Peer review of "Evaluation of CpG-ODN-Adjuvanted Toxoplasma gondii Virus-Like Particle Vaccine upon One, Two, and Three Immunizations"

_pharmaceutics, 2020, doi:10.3390/pharmaceutics12100989_

Round 1
Reviewer 1 Report
This manuscript demonstrates the use of a vaccine in combination with the adjuvant CpG-ODN to induce an effective immune response against T.gondii. The results are relevant and informative for vaccine development against this parasite.
However, I find several weaknesses that need to be improved.
1) Most figures do not show the amount of mice used and the number of experiments performed.
2) The analysis for the flow cytometry data is confusing. Better explanation is needed on how the cells were identified and the percentages quantified.
3) It looks like all the samples for the flow cytometry data were stimulated with T. gondii antigen. Were the unstimulated cells substracted from the stimulated cells? It is difficult to interpret the data without the unstimulated control.
In addition, I have several comments to improve the writing/understanding of the manuscript.
1) On the Introduction, it is important to mention that the CpG-ODN is a type C.
2) On line 188, the word propagation is not very appropriate, better use expansion.
3) The analysis of T cells is confusing, are these total CD4+ T cells, or T-gondii-specific? How do you know they were activated by the T. gondii antigen? I don't see any activation marker or cytokine levels.
4) The word "response" from the graphs on T cells and B cells should be removed. Instead % lymphocytes or leukocytes (or among CD4+) should be placed depending on how the analysis was done.
5) The figure legends for flow cytometry should mention how the cells were gated and identified.
6) The legend for figure 7 should not include result summaries, just how the experiment was done, number of mice, repeats and stats.
Best.
Author Response
Response to Reviewer 1 Comments
1) Most figures do not show the amount of mice used and the number of experiments performed.
Response: They have been added in the figure legends (lines: 160, 179, 206, 246, 271).
2) The analysis for the flow cytometry data is confusing. Better explanation is needed on how the cells were identified and the percentages quantified.
Response: Gating strategies for T and B cells were included in Fig 3 to illustrate cell identification and percentage quantification processes (Fig. 3, lines 197-202).
3) It looks like all the samples for the flow cytometry data were stimulated with T. gondii antigen. Were the unstimulated cells subtracted from the stimulated cells? It is difficult to interpret the data without the unstimulated control.
Response: CD4+ and CD8+ T cell responses were retained at similar levels regardless of antigen stimulation, as indicated in our previous studies (Pharmaceutics, 2019, 11, 342; doi:10.3390/pharmaceutics11070342; Frontiers in Immunology, https://doi.org/10.3389/fimmu.2018.03073). This information has been added (lines 131, 209). GC B cells and B cell populations were significantly increased after stimulation compared to those without stimulation (line 221). Data in Fig. 5 were acquired by subtracting the unstimulated cells from the stimulated cells. The figure legend has been revised accordingly (lines 228-229).
4) On the Introduction, it is important to mention that the CpG-ODN is a type C.
Response: It has been added in introduction section (lines: 67-68).
5) On line 188, the word propagation is not very appropriate, better use expansion.
Response: It has been revised (line: 189,194).
6) The analysis of T cells is confusing, are these total CD4+ T cells, or T-gondii-specific? How do you know they were activated by the T. gondii antigen? I don't see any activation marker or cytokine levels.
Response: Immune cells isolated from the spleen and MLN of mice were stimulated with or without T. gondii antigen for 2 hours to measure total CD4+ T cells. A gating strategy has been newly included in Fig 3 for detailed information.
7) The word "response" from the graphs on T cells and B cells should be removed. Instead % lymphocytes or leukocytes (or among CD4+) should be placed depending on how the analysis was done.
Response: They have been revised (Fig. 4, Fig. 5).
8) The figure legends for flow cytometry should mention how the cells were gated and identified.
Response: The Fig. 3 for gate strategy has been newly added (Fig. 3, lines 197-202).
9) The legend for figure 7 should not include result summaries, just how the experiment was done, number of mice, repeats and stats.
Response: It has been revised in Fig 8 (lines 293-297).
Reviewer 2 Report
The authors analyzed the CPG-DON adjuvant for Toxoplasma gondii VLP vaccine, and selected 3 boost manners. As expected, the three immunizations induced highest response for challenge. Comments listed:
- Immunization-times is a very important issue for vaccine development, as well as interval and doses, is it possible consider interval and doses factor, to design a comprehensive study for this vaccine. If just consider the adjuvant function, just one immunization manner is enough.
- Did author analyzed the unstimulated cell response? 2 hours is enough for stimulation? Is it still so high GCB level after 35 days post immunization? Please indicate flow gating strategy.
Author Response
Response to Reviewer 2 Comments
1) Immunization-times is a very important issue for vaccine development, as well as interval and doses, is it possible consider interval and doses factor, to design a comprehensive study for this vaccine. If just consider the adjuvant function, just one immunization manner is enough.
Response: In the current study, 4 weeks interval and 100 ug dosage were selected and the efficacies of advjuanted vaccines were assessed in mice following one, two, or three immunizations. We found that the degree of protection was proportional to the number of immunizations, and concluded that pairing adjuvant usage with multiple immunizations have a profound effect on reducing total cyst counts and body weight loss. Consistent with this notion, highest cyst counts which reflect the lowest level of protection, were observed in mice immunized once. To maximize the adjuvant effect upon one immunization as commented, a study considering both dose and immunization interval could be planned in the near future.
2) Did author analyzed the unstimulated cell response?
Response: Yes, we have compared cell responses with or without stimulation. We concluded that CD4+ and CD8+ T cells showed similar responses regardless of antigen stimulation as published in our previous studies (Pharmaceutics, 2019, 11, 342; doi:10.3390/pharmaceutics11070342; Frontiers in Immunology, https://doi.org/10.3389/fimmu.2018.03073). This information has been added (lines 131, 209). Contrastingly, B cells were significantly increased post-stimulation in comparison to the unstimulated control (line 221). Both Fig. 5 and the figure legends have been revised by subtracting the unstimulated cells from the stimulated cells (lines 228-229).
3) 2 hours is enough for stimulation?
Response: Yes. We have compared data after 2, 3, and 5 hours of stimulations, and concluded that 2 hours is enough.
4) Is it still so high GCB level after 35 days post immunization?
Response: In the current study, GCB level has been analyzed at 35 days after challenge infection as described in the manuscript. The levels of GCB post-immunization will be conducted in the near future.
5) Please indicate flow gating strategy.
Response: Gating strategy has been newly added as Fig. 3 along with the figure legends (Fig. 3, lines 197-202).
Round 2
Reviewer 1 Report
Thanks for including the changes I suggested, those have improved the quality of the manuscript significantly.
The only concern I have now is that you state that you analyzed antigen-specific CD4+ and CD8+ T cell responses but that is not a correct statement. Since you did not analyze T cell activation, antigen-specific T cell receptors, nor cytokine production on the stimulated cells, the results you are showing only demonstrate CD4+ or CD8+ T cell frequency after immunization. Also, since the in vitro culture stimulation was only 2hrs, that is not enough time for T cells to proliferate. It is still a relevant analysis, since you quantified T cells after the immunizations. Therefore, everywhere you mention antigen-specific T cell responses must be changed to frequency, percentage or expansion (in cases where you actually saw that) of total CD4+ and CD8+ T cells. Therefore, change:
line 29: remove responses
line 185: change to "induce CD4+ and CD8+ T cell expansion"
line 189: change to "CD4+ and CD8+ T cell frequency"
line 192: change to "CD4+ and CD8+ T cell frequencies"
line 205: change to "enhances CD4+ and CD8+ T cell frequencies"
Best to you!
Author Response
Response to Reviewer 1
Thanks for including the changes I suggested, those have improved the quality of the manuscript significantly.
The only concern I have now is that you state that you analyzed antigen-specific CD4+ and CD8+ T cell responses but that is not a correct statement. Since you did not analyze T cell activation, antigen-specific T cell receptors, nor cytokine production on the stimulated cells, the results you are showing only demonstrate CD4+ or CD8+ T cell frequency after immunization. Also, since the in vitro culture stimulation was only 2hrs, that is not enough time for T cells to proliferate. It is still a relevant analysis, since you quantified T cells after the immunizations. Therefore, everywhere you mention antigen-specific T cell responses must be changed to frequency, percentage or expansion (in cases where you actually saw that) of total CD4+ and CD8+ T cells. Therefore, change:
line 29: remove responses
Response: It has been removed (line 29).
line 185: change to "induce CD4+ and CD8+ T cell expansion"
Response: It has been corrected as indicated (lines 185-186).
line 189: change to "CD4+ and CD8+ T cell frequency"
Response: It has been corrected as indicated (line 189).
line 192: change to "CD4+ and CD8+ T cell frequencies"
Response: It has been corrected (line 192).
line 205: change to "enhances CD4+ and CD8+ T cell frequencies"
Response: It has been corrected (lines 204-205).
Reviewer 2 Report
Thank you so much for the detailed response.
Author Response
Response to Reviewer 2 Comments
Thank you so much for the detailed response.
Response: Thank you very much!